# The acceptability and uptake of smartphone tracking for COVID-19 in Australia

**Paul M. Garrett** [ID][1]*, **Joshua P. White** [ID][1], **Stephan Lewandowsky**[2], **Yoshihisa Kashima**[1], **Andrew Perfors**[1], **Daniel R. Little** [ID][1], **Nic Geard**[3,4], **Lewis Mitchell** [ID][5], **Martin Tomko** [ID][6], **Simon Dennis**[1,7]

1 Melbourne School of Psychological Sciences, The University of Melbourne, Melbourne, Victoria, Australia, 2 School of Psychological Science, The University of Bristol, Bristol, United Kingdom, 3 School of Computing and Information Systems, The University of Melbourne, Melbourne, Victoria, Australia, 4 Peter Doherty Institute for Infection and Immunity, Melbourne, Victoria, Australia, 5 School of Mathematical Sciences, The University of Adelaide, Adelaide, South Australia, Australia, 6 School of Electrical, Mechanical and Infrastructure Engineering, The University of Melbourne, Melbourne, Victoria, Australia, 7 Unforgettable Research Services, Melbourne, Victoria, Australia

☯ These authors contributed equally to this work.
* paul.garrett@unimelb.edu.au

**Data Availability Statement:** The data files underlying the results presented in the study are available through the Open Science Framework at osf.io/sw7rq.

## Abstract

In response to the COVID-19 pandemic, many Governments are instituting mobile tracking technologies to perform rapid contact tracing. However, these technologies are only effective if the public is willing to use them, implying that their perceived public health benefits must outweigh personal concerns over privacy and security. The Australian federal government recently launched the 'COVIDSafe' app, designed to anonymously register nearby contacts. If a contact later identifies as infected with COVID-19, health department officials can rapidly followup with their registered contacts to stop the virus' spread. The current study assessed attitudes towards three tracking technologies (telecommunication network tracking, a government app, and Apple and Google's Bluetooth exposure notification system) in two representative samples of the Australian public prior to the launch of COVID-Safe. We compared these attitudes to usage of the COVIDSafe app after its launch in a further two representative samples of the Australian public. Using Bayesian methods, we find widespread acceptance for all tracking technologies, however, observe a large intention-behaviour gap between people's stated attitudes and actual uptake of the COVIDSafe app. We consider the policy implications of these results for Australia and the world at large.

## Introduction

The COVID-19 pandemic caused by the severe acute respiratory coronavirus 2 (SARS-CoV-2) disease has changed how Australians live, work and socialise. In the absence of a vaccine or treatment, behavioral measures such as restricting public gatherings and physical distancing [1], mask wearing [2], lockdown policies, and hand-washing [3] have been the only means available to slow the spread of the virus (Fig 1). The easing of restrictions and reduction of compliance preempted a 'second wave' of infections, shown in Fig 1 [4]. The highly

**Funding:** This work was supported by anonymous philanthropic funding to the Peter Doherty Institute for Infection and Immunity, awarded to S.D. (https://www.doherty.edu.au). The funder provided support in the form of salaries for the author P.G., and in the payment of participants, but did not have any additional role in the study design, data collection and analysis, decision to publish, or preparation of the manuscript. The specific roles of these authors are articulated in the 'author contributions' section.

**Competing interests:** The authors have read the journal's policy and have the following competing interests: author S.D. is the CEO of Unforgettable Research Services Pty Ltd (URS) that specializes in providing privacy-preserving experience-sampling collection and analysis services. This does not alter our adherence to PLOS ONE policies on sharing data and materials. There are no patents, products in development or marketed products associated with this research to declare.

transmittable [5] and often asymptomatic nature of this virus [6] suggests a benefit for technological solutions to curb its spread [7]. Smartphone tracking technologies offer one such solution.

## Smartphone tracking technologies

Smartphones tracking technologies use GPS, telecommunication, or Bluetooth data to create a list of contacts with whom a user may have been co-located [7]. This contact information is either stored locally on the phone (decentralized) or on a centralised server. Partially centralized systems also exist that allow users to decide when data is uploaded to a central sever. If a registered contact later tests positive with COVID-19, the user can be instantly notified, allowing them to self-isolate and get tested, thus helping to slow the virus' spread [8]. A description of these technologies and storage options are provided in Fig 2.

People engage in a "privacy calculus" when balancing the benefits (public health) and consequences (surveillance creep) of disclosing sensitive data [13]. For mobile tracking technologies to be effective, their benefits must outweigh concerns over privacy, security, and risk of harm [14]. In a recent survey of 100 Australians, Kininmonth et al., (2018) [15] found Government surveillance was acceptable when conducted by a "trusted" source for "necessary" reasons, meaning perceptions about the risk from COVID-19 and perception about those collecting the data (e.g., Governments or corporations), will impact public acceptance for COVID-19 tracking technologies.

Newly published findings by Wnuk et al. (2020) on COVID-19 tracking acceptance among Polish adults exemplifies how the perceived threat of COVID-19 influences public acceptance of tracking technologies. Regression analysis revealed perceived COVID-19 threat, lack of control over one's life due to COVID-19, and ideological beliefs were all predictive of COVID-19

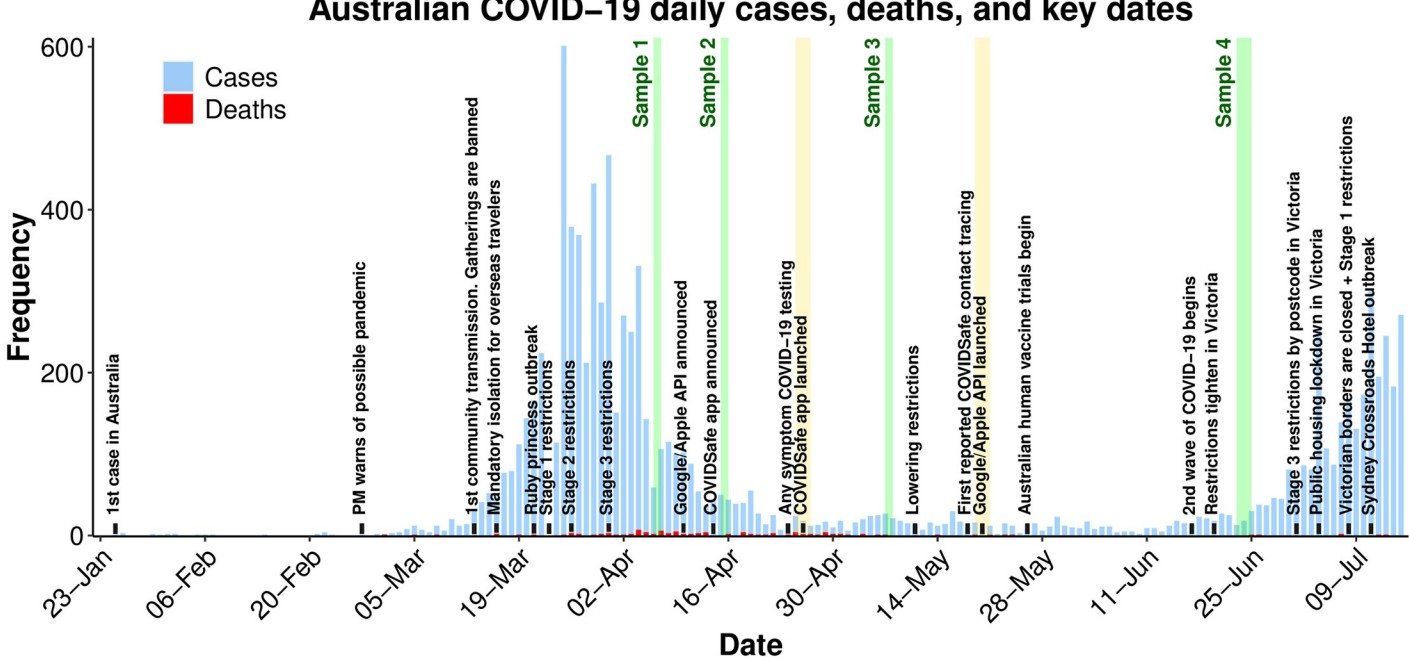

**Fig 1. Australian COVID-19 cases, deaths, and key-dates.** COVID-19 daily cases (blue), deaths (red), and key policy decisions (text) in Australia during the COVID-19 pandemic within the period January 23rd–July 14th, 2020. Collection dates of the current study are highlighted in green and the introduction of key tracking technologies are highlighted in yellow. A record of news sources for this fig are included on our OSF page, osf.io/sw7rq.

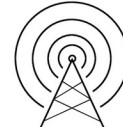

**Telecommunication Network Tracking**

- 50 meters – 2KM radius[9].

- May identify rough locations: work, home or school.

- Access highly restricted.

- Does not require the users consent, but does not work if phone is off/in airplane mode.

- Works on any phone (not just smartphones).

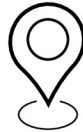

**GPS Tracking**

- Precise to within 5 meters[10].

- Performs poorly in built-up areas and indoors (where the virus is likely to spread).

- Geolocation tracking risks privacy & security.

- Only works on GPS enabled smartphones.

- Voluntary. Can be turned off.

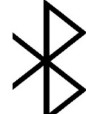

**Bluetooth Tracking**

- Usually a 10 meter radius[11,12].

- Obstructed by objects.

- Records who you have been in proximity to, not locations.

- Anonymized IDs; low threat to privacy and security.

- Only works on Bluetooth enabled phones.

- Voluntary. Can be turned off.

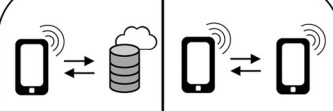

**Centralized vs Decentralized Storage**

- **Centralized**: data stored on external servers. Contact tracers access data & notify contacts[8].

- **Decentralized**: Data stored on the phone. Notifications are coordinated phone-to-phone.

- **Part-centralized** data voluntarily uploaded to a server if infected. Contact tracers notify contacts.

**Fig 2. Mobile tracking technologies.** Infographic highlighting the differences between telecommunication [9], GPS [10], and Bluetooth tracking [11, 12], and the distinction between centralized and decentralized data storage [8].

tracking acceptance. Similar findings have also been observed in the United Kingdom [16] and among young adults in Taiwan [17]. Understanding the complex interplay between perceptions of the virus' threat, and the personal risks and public health benefits that accompany COVID-19 tracking technologies, is crucial for policy decision makers who wish to introduce effective technological solutions for curtailing the spread of the SARS-CoV-2 virus (for further social and behaviour solutions, see [1]).

The current research aims to extend our theoretical understanding about the privacy calculus and the shifting of public attitudes during important real-world situations. Our investigation will focus on a limited set of three real-world COVID-19 tracking technologies that can be rapidly implemented by governments or corporations, and that use a readily available and widely distributed piece of hardware: mobile smartphones. To these ends, we focus on the following COVID-19 tracking technologies: telecommunication network tracking (e.g., Taiwan's electronic fence [18]), a Government smartphone app (e.g., Australia's COVIDSafe [19]), and Apple and Google's exposure notification (EN) Bluetooth system (e.g., Germany's Corona Warn-App [20]). Practically, this research aims to inform policy makers of the necessary conditions for public acceptance of mobile tracking technologies during the COVID-19 pandemic. Although our investigation focuses exclusively on Australia, our results may prove informative for other countries considering the introduction of mobile contract tracing technologies.

## Mobile tracking in Australia

Telecommunication tracking has already been used by the Australian Government to create anonymized social mobility reports but not to individuate users or locate them relative to each other [21]. Theoretically, the Government could access this data to locate users without their consent under the auspices of national security [22], however, such access is highly

restricted and data breaches are a criminal offense. This preexisting tracking infrastructure is further hindered by the precision of the technology, becoming less accurate where network towers are sparsely populated [9]. Practically, this means many individuals could be falsely identified as 'relatively close' to an infected individual due to poor tracking precision [23], ultimately harming acceptance and trust in the technology. In an attempt to decrease the chance for false-alarms, the Australian Government has instead pursued a partially-centralized Bluetooth government app.

On April 26th, 2020, the Australian Government introduced its COVID-19 mobile tracking app called COVIDSafe [19]. The app uses Bluetooth technology to create an anonymous register of nearby app-users stored locally on their phone [24]. If an app-user voluntarily identifies as infected, their COVIDSafe contacts are called by members of the health department and are encouraged to get tested and stay home. The registry is then sent to manual contact tracers working in the health department to contact potentially infected individuals who were not using the app.

The COVIDSafe app implements several measures to ensure data privacy and security. Contact registry identifications are randomised every two-hours, registry contacts are deleted after 21 days [25], and the app's client-side code (i.e., not the server code where privacy measures are implemented) is freely available [26]. Furthermore, the newly legislated Privacy Amendment Act 2020 mandates that COVIDSafe data must be destroyed once the app is no longer effective, and that individuals cannot be forced to download the app or be discriminated against for not doing so [27].

As of July 7th, the app was downloaded 6.5 million times (approximately half the number targeted by the Government) [28]. By late August, the app had only traced six individuals not already identified by manual tracing efforts [29] prompting some to question whether the lack of effectiveness is related to insufficient uptake. If so, it is important to understand what factors underlie Australians' willingness to download and use the COVIDSafe app, and if other options would be more effective.

## The current study

Prior to the release of the COVIDSafe app, we asked two representative samples of the Australian public about their attitudes to three hypothetical smartphone tracking scenarios—a Government app, telecommunication tracking, and the Apple and Google exposure notification 'EN' Bluetooth system. The EN system launched on May 20th 2020 [30], is functionally identical to COVIDSafe except that data is stored in a decentralized system and contact registries can be shared between any two apps that use the EN system [31]. We then compared attitudes to these hypothetical scenarios to a further two representative samples that reported on their usage of the newly released COVIDSafe app. This allowed us to assess the changing attitudes towards tracking technologies conditioned by both the state of the pandemic, and the changing literacy about tracking technologies in the population, and to determine how public attitudes inform the uptake of COVID-19 tracking technologies. As the reader will soon see, the tracking scenarios presented to participants varied on several dimensions (e.g., voluntary vs mandatory tracking, centralized vs decentralized storage, government vs corporate data ownership). We did this to assess public attitudes to real-world tracking policies that were under consideration in Australia, so as to better inform policy decision makers in their response to the COVID-19 pandemic. We did not have specific hypotheses regarding public acceptance of these scenarios, or public uptake of the COVIDSafe app. Instead, we directly assessed public attitudes and reported a descriptive account of our findings.

## Materials and methods

### Overview

We present four survey samples collected during the COVID-19 pandemic in Australia between April 6th and June 23rd 2020 (see Fig 1). The surveys examined attitudes towards smartphone tracking technologies, in addition to a range of constructs such as world views, trust in government, and the impact and perception of COVID-19.

Sample 1 examined two hypothetical tracking scenarios: One a voluntary government app that notified users if they were in contact with an infected individual and shared contact registries with the health department ("Gov App" scenario), and one in which all mobile users were tracked through the telecommunication network through which the government could issue quarantine orders ("Telecommunication" scenario). The Gov App scenario was intended to gauge attitudes towards the soon-to-be released COVIDSafe app; however, it was necessarily vague on how the app would work as these details had not been announced at the time.

Sample 2 included an additional "Bluetooth" scenario, describing the EN system proposed by Apple and Google. Phones exchanged contact information and notified users if a nearby individual later identifies as infected; the government was never informed of these individuals.

Samples 3 and 4 focused exclusively on a real-world tracking scenario, the COVIDSafe app. COVIDSafe was described as functionally identical to the "Bluetooth" scenario, except that contact information was accessible by the health department to enhance manual contact tracing efforts and all data was collected and secured by the Government.

### Ethics statement

This study received ethics approval from the University of Melbourne's psychology health and applied sciences human ethics sub-committee, approval number 1955555. Participants gave informed consent and were debriefed at the end of each survey. Information and debriefing sheets are included in each Qualtrics survey, versions of which can be found at osf.io/sw7rq.

### Participants

Participants were a representative sample of the Australian public stratified by gender, age, and state per the 2016 census [32] and obtained through the data sampling platform Dynata. Collection dates, sample size, and cumulative COVID-19 cases and deaths are summarized in Table 1. Participants were aged 18 years or older and completed a 10-minute (Sample 1) or 15-minute (Samples 2–4) online survey for which they were reimbursed in the form of gift cards, points programs, or charitable contributions as per their agreement with Dynata.

### Design and procedure

Surveys can be downloaded in full at osf.io/sw7rq. Fig 3 illustrates the survey design at each stage of data collection. Plain language statements, consent, comprehension checks directly following each scenario, and free-text responses at the end of the survey are not shown.

**Table 1. Sample information.** Collection dates, sample size, and cumulative COVID-19 cases and deaths in Australia at the time of data collection.

|  | Date | *N* | Cumulative Cases | Cumulative Deaths |
|---|---|---|---|---|
| Sample 1 | April 6th 2020 | 1275 | 5744 | 36 |
| Sample 2 | April 15th 2020 | 1777 | 6416 | 61 |
| Sample 3 | May 7th 2020 | 597 | 6738 | 88 |
| Sample 4 | June 23rd 2020 | 596 | 7474 | 102 |

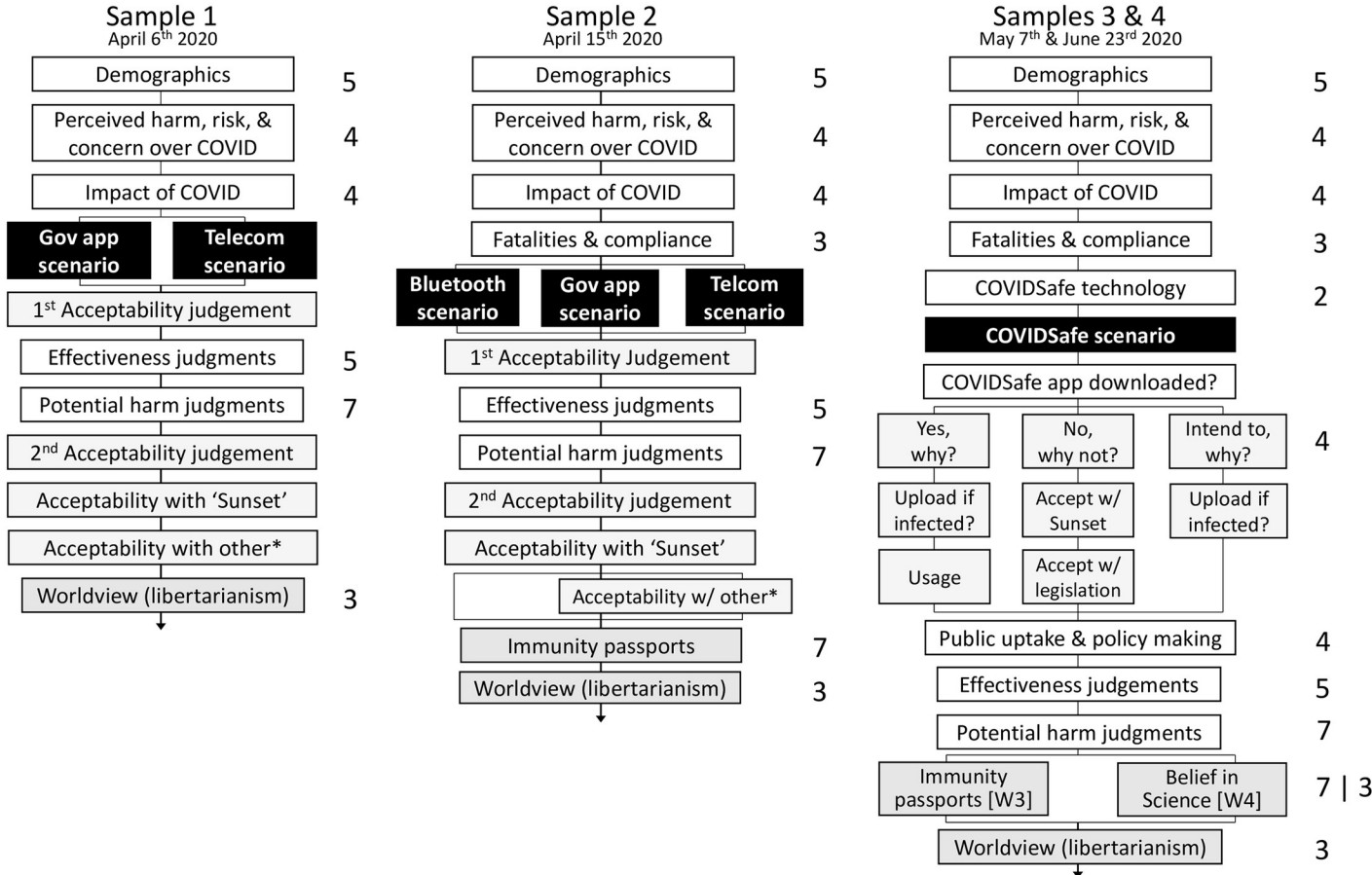

**Fig 3. Survey design for Samples 1–4.** White boxes depict a block of questions with the number of items displayed on the right. Black boxes display the scenario to which participants were randomly assigned (between-subjects design) and gray boxes illustrate judgments of tracking acceptability. 'Acceptability with other*' included a local phone data-storage option for the government app scenario and the ability to opt-out of tracking in the telecommunication scenario. Items not included in the results of this paper are shaded gray.

Immunity passports (Samples 2 and 3), worldview items, and beliefs in science (Sample 4) will not be discussed in the current paper.

After providing consent and demographic information, participants were asked about how they perceived the risk of COVID-19 (Risk items summarized in Table 2). Responses were

**Table 2. Perceived risk.** Items assessing the perceived risk from COVID-19.

| Item | Question | Label |
|---|---|---|
| Risk 1 | How severe do you think novel coronavirus (COVID-19) will be for the general population? | General harm |
| Risk 2 | How harmful would it be for your health if you were to become infected COVID-19? | Personal harm |
| Risk 3 | How concerned are you that you might become infected with COVID-19? | Concern self |
| Risk 4 | How concerned are you that somebody you know might become infected with COVID-19? | Concern others |
| Impact 1 | Have you ever tested positive to COVID-19? | Positive self |
| Impact 2 | Has somebody you know ever tested positive to COVID-19? | Positive other |
| Impact 3 | How many days, if any, have you been in quarantine or self-isolation? | Lockdown days |
| Impact 4 | Have you temporarily or permanently lost your job as a consequence of the COVID-19 pandemic? | Job loss |

made on a 5-point scale, where increasing values were associated with greater endorsement of the issue, (e.g., 1 = Not at all, 5 = Extremely). Participants were then asked about the impact from COVID-19 (see impact items, Table 2) and were asked to estimate national fatalities and policy compliance (Samples 2–4), before being randomly assigned to read a single tracking scenario description. In Samples 3 and 4, participants were additionally asked what technology, (e.g., Bluetooth, GPS, telecommunication network), COVIDSafe used prior to reading the scenario description. Following each description, participants responded to a comprehension question; participants who did not identify the scenario from among three foils were excluded at analysis.

The government app scenario was described as:

"The COVID-19 pandemic has rapidly become a worldwide threat. Containing the virus' spread is essential to minimize the impact on the healthcare system, the economy, and save many lives. The Australian Government might consider using smartphone tracking data to identify and contact those who may have been exposed to people with COVID-19. This would help reduce community spread by identifying those most at risk and allowing health services to be appropriately targeted. **Only people that downloaded a government app and agreed to be tracked and contacted would be included in the project**. The more people that download and use this app the more effectively the Government would be able to contain the spread of COVID-19. Data would be stored in an encrypted format on a secure server accessible only to the Australian Government. **Data would only be used to contact those who might have been exposed to COVID-19**." Bold text is displayed as it was in the survey and the Chinese translation used during the survey is provided in the supporting information.

The telecommunication tracking scenario was described as:

"The COVID-19 pandemic has rapidly become a worldwide threat. Containing the virus' spread is essential to minimize the impact on the healthcare system, the economy, and save many lives. The Australian Government might consider using phone tracking data supplied by telecommunication companies to identify and contact those who may have been exposed to people with COVID-19. This would help reduce community spread by identifying those most at risk and allowing health services to be appropriately targeted. **All people using a mobile phone would be included in the project, with no possibility to opt-out**. Data would be stored in an encrypted format on a secure server accessible only **to the Australian Government who may use the data to locate people who were violating lockdown orders and enforce them with fines and arrests where necessary**. Data would also be used to inform the appropriate public health response and to contact those who might have been exposed to COVID-19, and individual quarantine orders could be made on the basis of this data."

The Bluetooth (Apple and Google exposure notification system) scenario was described as:

"The COVID-19 pandemic has rapidly become a worldwide threat. Containing the virus' spread is essential to minimize the impact on the healthcare system, the economy, and save many lives. **Apple and Google have proposed adding a contact tracing capability to existing smartphones to help inform people if they have been exposed to others with COVID-19**. This would help reduce community spread of COVID-19 by allowing people to voluntarily self-isolate. When two people are near each other, their phones would

connect via Bluetooth. If a person is later identified as being infected, the people they have been in close proximity to are then notified **without the government knowing who they are**. **The use of this contact tracing capability would be completely voluntary**. **People who are notified would not be informed who had tested positive**."

And the COVIDSafe app scenario was described as:

"The COVID-19 pandemic has rapidly become a worldwide threat. Containing the virus' spread is essential to minimise the impact on the healthcare system, the economy, and save many lives. The Australian Government has recently released the COVIDSafe smartphone app to help identify and contact those who may have been exposed to people with COVID-19. **The use of this app is completely voluntary, but the government has explicitly stated that easing social distancing restrictions depends at least in part on the degree of community uptake of this voluntary app**. This is because the more people that download and use this app the more effectively it will help to contain the spread of COVID-19. **The app works with Bluetooth and no location data is collected**: when two people are near each other, their phones connect and keep a record of all these connections. If a person is later identified as being infected, that person may **voluntarily upload their Bluetooth contacts to a secure server accessible only to the Health Department of the Australian Government**. **This data would only be used by the Health Department of the Australian Government to contact those who might have been exposed to COVID-19**."

Tracking acceptability was assessed directly after the scenario description (1st acceptability) and after responding to items on the benefits and harm posed by the scenario (2nd acceptability). In the government app scenario, participants were asked if they "would download and use" the app, in the telecommunication scenario if "the use of tracking data in this scenario is acceptable", and in the Bluetooth scenario if they "would use" the technology. Items assessing the benefits and harm posed by each scenario are summarized in Table 3. Responses were made on a 6-point scale, (e.g., 1 = Not at all, 6 = Extremely). Reverse scored items are denoted by [R] in the table.

In samples 1 and 2, the second acceptability judgement was followed by two items only if the described scenario was deemed unacceptable. The first item asked if tracking would be acceptable under a sunset clause where data was deleted after 6 months. The second item

**Table 3. Perceived benefits and harms.** Items assessing the benefits and harm arising from smartphone tracking. 'The Government' was replaced by 'Apple and Google' in the Bluetooth scenario.

| Item | Question | Label |
|------|----------|-------|
| Bfit 1 | How confident are you that the described scenario would reduce your likelihood of contracting COVID-19? | Reduce contraction |
| Bfit 2 | How confident are you that the described scenario would help you resume your normal activities more rapidly? | Resume activity |
| Bfit 3 | How confident are you that the described scenario would reduce the spread of COVID-19? | Reduce spread |
| Harm 1 | How difficult is it for people to decline participation? | Difficult to decline [R] |
| Harm 2 | To what extent do people have ongoing control of their data? | Ongoing control |
| Harm 3 | How sensitive is the data being collected? | Data sensitivity |
| Harm 4 | How serious is the risk of harm from the proposed scenario? | Risk from tracking |
| Harm 5 | How secure is the data that would be collected? | Data security [R] |
| Harm 6 | To what extent is the Government [Apple/Google] only collecting the data necessary to achieve the purposes of the policy? | Data necessary |
| Harm 7 | How much do you trust the Government [Apple/Google] to use the tracking data only to deal with the COVID-19 pandemic? | Trust intentions |
| Harm 8 | How much do you trust the Government [Apple/Google] to be able to ensure the privacy of each individual? | Trust privacy |

**Table 4. COVIDSafe questions.** COVIDSafe follow-up questions conditioned on if the app was downloaded. Response options shown in brackets.

| Downloaded | Follow up questions | Label |
|---|---|---|
| Yes | When did you download the app? [The day/week/anytime-after the app was released] | When download |
| Yes | Do you have the COVIDSafe app downloaded, registered and have Bluetooth switched on? [Yes/No] | Effective use |
| Yes* | Do you leave your phone screen unlocked, Bluetooth on, and COVIDSafe open on the front screen? [Yes/No] | Effective use iOS |
| Yes | If you test positive to COVID-19, will you upload your data for manual contact tracing? [Yes/No] | Upload data |
| Yes** | Why did you download COVIDSafe? | Download why |
| Yes** | Who convinced you to download the app? | Download who |
| No** | Why won't you download the COVIDSafe app? | Not download why |
| No** | Who convinced you to not download the COVIDSafe app? | Not download who |
| Intend to | If you test positive to COVID-19, will you upload your data for manual contact tracing? [Yes/No] | Upload data |
| Intend to | Why haven't you downloaded the COVIDSafe app yet? | Waiting to download |
| All | Approximately what percentage of the people you know have downloaded the COVIDSafe app? [0–100] | Friends downloaded |
| All | If an insufficient number of people download the app, should the Government make it mandatory by issuing fines? [Yes/No] | Mandatory download |

*Only presented to Apple iOS users; prior to the June 19th update of COVIDSafe the app only worked on Apple iOS devices under the described conditions.

**Denotes items with rank-ordered responses.

asked if tracking would be acceptable if participants could opt-out of tracking (telecommunication scenario), or if data was only stored locally on the phone rather than on a government server (government app scenario).

In Samples 3 and 4, participants were asked if they "have downloaded the COVIDSafe app" and could respond "yes", "no", or that they "intend to download the app in the future". Then, participants were presented follow-up questions probing why they did or did not download COVIDSafe. These items are summarized in Table 4 and their response options are summarized in square brackets following each question. Following these questions, participants responded to items probing the benefits and harm posed by the COVIDSafe scenario (Table 3).

Assessments of acceptability were then followed by items probing attitudes to introducing immunity passports (Samples 2 and 3), beliefs in the benefits of science (Sample 4), and political worldviews. Beliefs in the benefits of science were probed by three items on a 0–10 sliding scale, (e.g., 0 = Completely agree, 10 = Completely disagree). These items will not be reported in the current paper.

In Sample 4, participants were asked three final questions. One item measured attitudes to downloading another country's COVID-19 tracking app and a second item measured attitudes on requiring travellers to Australia to download COVIDSafe. A final item asked how participants viewed the Government's response to the COVID-19 pandemic. Items were responded to using a seven-point scale, (e.g., 1 = strongly approve, 7 = strongly disapprove).

## Data analysis

Bayesian ordinal probit regressions were used to compare Likert-style responses using the MCMCoprobit and HPDinterval functions in R packages MCMCpack [33] and Coda [34], respectively. This method assumes that there are latent normally distributed continuous variables that underlie ordinal responses. These latent variables are then segmented into ordinal Likert responses by $C - 1$ (number of response options—1) thresholds. To set the location of the underlying latent variable and make the model identifiable, the lowest threshold parameter is fixed at zero [35] and all other thresholds are estimated. Similar items and different scenario conditions were modelled together to ensure consistent threshold parameters across Likert

items and across scenarios (see [36] for model details). For binomial distributions (i.e. proportions of 'yes', from a yes or no response variable), Bayesian credible intervals were calculated using the bayes.prop.test function from the BayesianFirstAid package [37].

These Bayesian methods sample a posterior distribution of plausible means (the probability that, given our data, the true population mean is 'x'), by weighing the likelihood of a given observation against its prior probability of occurring in the sample. Under parametric assumptions, these posterior distributions act to constrain the effect of outliers in the tails of the sampled data, and allow the highest region of data density—credible regions of the data distribution—to inform policy decisions. Practically, this means instead of testing a threshold of significance (like p-values or Bayes factors), we may instead directly compare the 95% credible regions of the data distributions to determine if they overlap or not.

The MCMCoprobit function was run with one chain of 20,000 Markov Chain Monte Carlo (MCMC) iterations (including 1000 burn-ins) per variable, and a tuning parameter of 0.3 (corresponding to the size of the Metropolis-Hastings step). Default priors were used for all parameters (i.e., the distributional means and the cutpoints), corresponding to an improper uniform prior (i.e., over real numbers between the range of -inf and inf) for both the latent variable means and the threshold parameters. The bayes.prop.test function was run with one chain with 20,000 MCMC iterations (including 1000 burn-ins). Default priors were again used: a beta distribution with parameters of $\alpha = 1$ and $\beta = 1$, corresponding to a uniform prior over the unit interval. Ninety-five percent highest posterior density intervals (HDIs) were estimated on the resulting posterior samples. Anonymized data and analysis code for this project is available at osf.io/sw7rq. Unless specified otherwise, results for the same scenarios will be collapsed across Samples 1 and 2, and results regarding COVIDSafe will be collapsed across Samples 3 and 4.

## Results

### Data preparation and demographics

Participants who did not reside in Australia, were under the age of 18, failed to pass the comprehension check, or did not complete the survey were removed from each sample. Removals, final sample sizes and demographic information is provided in Table 5. Samples were representative per the 2016 census, except for education, which was biased towards University Graduates. Across Samples 2–4, 93% of participants owned a smartphone (this question was not included in Sample 1).

### Perceived risk from COVID-19

Fig 4 displays the mean ordinal regression posterior distributions and associated likert-style responses for items querying people's perceived risk from COVID-19 in each Sample of data collection. Risk items are described in Table 2. Error bars display the 95% HDI and black horizontal lines illustrate differences where HDIs do not overlap. Posterior means decreased over time from very-to-somewhat for perceived general harm, concern for self, and personal harm items. Over time, participants consistently reported being 'very' concern for others.

### Perceived benefits from tracking

Fig 5 displays the mean ordinal regression posterior distributions and associated likert-style responses for items querying people's perceived benefits from tracking in each scenario. Benefit items are described in Table 3. Telecommunication was perceived as most beneficial with perceptions indicating a moderate reduction in contraction and spread of the virus, and

**Table 5. Demographics.** Sample size and demographics for data collection Samples 1–4.

| | | Sample 1 | Sample 2 | Sample 3 | Sample 4 |
|---|---|---|---|---|---|
| Initial Sample | | 1275 | 1777 | 597 | 596 |
| Removals | Not in Australia | 60 | 103 | 31 | 36 |
| | Under 18 years | 8 | 8 | 2 | 1 |
| | Comprehension check | 252 | 361 | 69 | 85 |
| | Incomplete | 134 | 136 | 46 | 45 |
| **Final Sample** | | **821** | **1169** | **449** | **429** |
| Gender (%) | Men | 52% | 50% | 49% | 50% |
| | Women | 48% | 49% | 51% | 49% |
| | Other | 0.1% | 0.2% | 0.2% | 0.2% |
| | Not provided | 0% | 0.3% | 0% | 0.2% |
| Age (years) | Mean | 49 | 48 | 47 | 47 |
| | Std. Dev | 17 | 17 | 17 | 17 |
| Max. Education (%) | Less-than High School | 9% | 9% | 11% | 8% |
| | Graduated High School | 38% | 37% | 40% | 34% |
| | Graduate University | 52% | 54% | 49% | 57% |

moderate ability to aid a return to normal activities. Bluetooth and government app scenarios were 'a bit' effective and did not vary across benefit items. COVIDSafe was perceived to be the least beneficial technology.

## Perceptions of tracking technologies

Fig 6 displays the mean ordinal regression posterior distributions and associated likert-style responses for items querying people's perceptions of the tracking technologies. Voluntary technologies were perceived as similarly easy to decline. Apple and Google's Bluetooth technology was perceived to collect the most non-essential data, and to have less-trustworthy intentions, and poorer data privacy and security than the Government alternatives. COVIDSafe was perceived as collecting the least sensitive data, as being the least risky technology, and as having the most ongoing control. Usability (not in Fig 6) for the app technologies—COVIDSafe, Bluetooth, and government app—was consistently reported as 'moderate'.

## Acceptability of tracking technologies

Fig 7 shows acceptability ratings and COVIDSafe downloads under varying conditions. Baseline acceptability ratings were measured after responding to the tracking effectiveness items (2nd acceptability, Fig 3). The remaining items show the subsequent increase in acceptability under a sunset clause and with an opt-out or local storage option. Intentions to download under appropriate legislation, and intentions to download in the future (regardless of legislation) are similarly displayed for COVIDSafe. Acceptability was high (62–70%) and did not differ meaningfully across the scenarios, and increased with additional privacy options in the telecommunication and government app scenarios (up to 87% and 70%, respectively). COVIDSafe downloads (44%) and intentions to download (58%) were lower than predicted by acceptability for the Government app, indicating an intention-behavior gap.

Fig 8 shows acceptability ratings for the three tracking scenarios over time and future intentions to download the COVIDSafe app (so as to be comparable to 'intentions' measured in Samples 1 and 2). Posterior HDIs indicate similar intentions to use the hypothetical

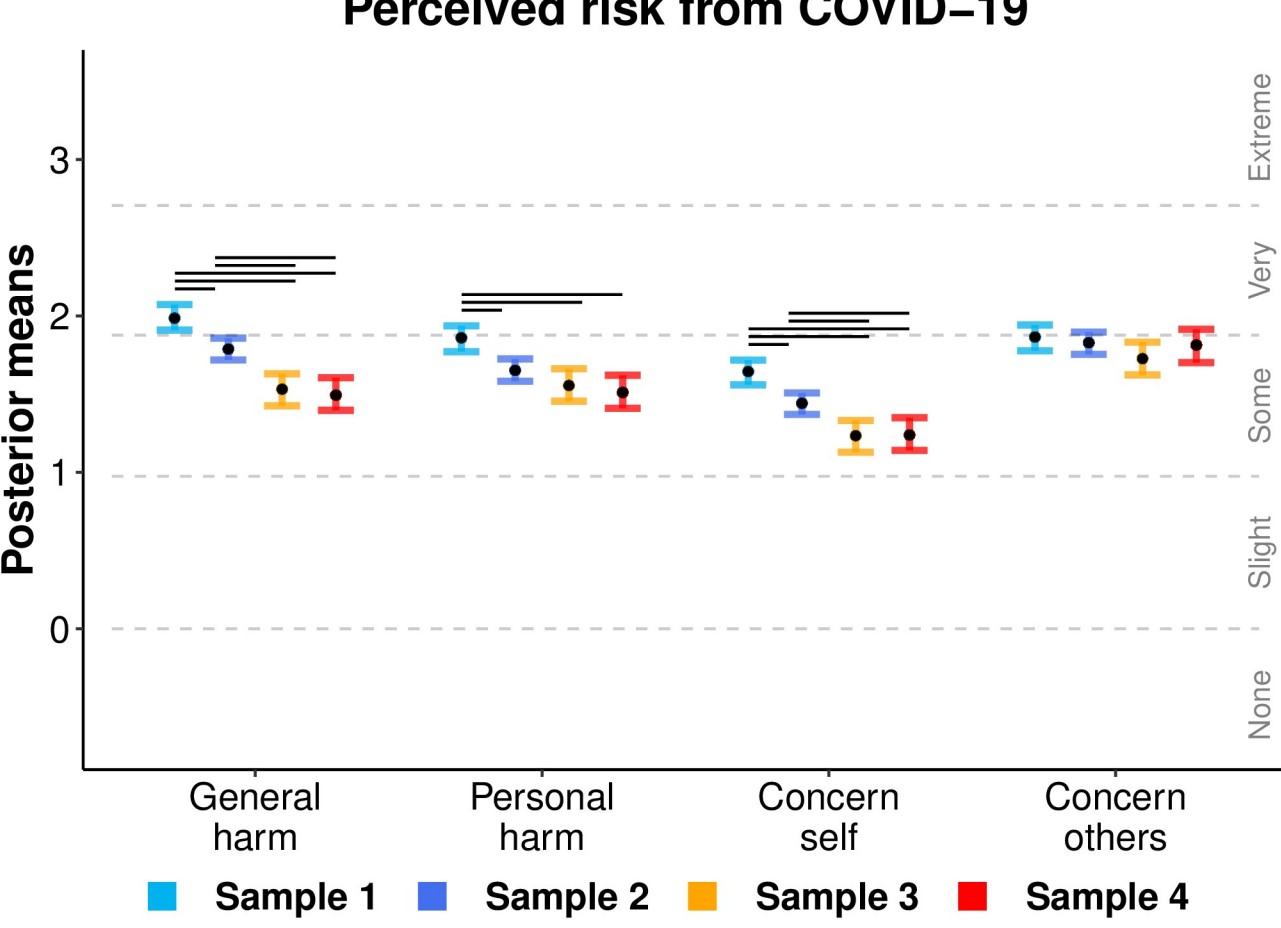

**Fig 4. Ordinal regression mean posterior distributions for items assessing the perceived risk from COVID-19 for each sample.** Black points display mean point estimates and coloured error bars display the 95% highest posterior density interval thereof. Dotted lines indicate the ordinal regression threshold parameters which separate the continuous latent variables into the ordinal response categories made by participants ('None' to 'Extremely'). Non-overlapping intervals (within items) are denoted by horizontal bars above each comparison.

government app and to download the COVIDSafe app over time, and a decline in acceptability for telecommunication tracking between Samples 1 and 2.

## COVIDSafe: Usage and perceptions

The following results were obtained in Samples 3 and 4, and focus exclusively on attitudes towards and real-world usage of the COVIDSafe app, (e.g., if, when, and why people downloaded the app). These results provide a snapshot of how attitudes and intentions (collected in Samples 1 and 2) differ to real-world behaviours (Samples 3 and 4), informing the previously measured intention-behavior gap.

**Usage and effectiveness.**   By Sample 4, most COVIDSafe users downloaded the app within the first day (29%) or week (57%) after its launch. The app's effective usage rate (i.e., percentage of people who correctly use the app) was high across Samples 3 and 4; 87% of users had the app installed, were registered and kept Bluetooth on when in public. Independently, app registration and installation was recorded as 94%, and Bluetooth usage was recorded at 71% 'at all times' and at 89% 'when in public'. Ninety-eight percent users responded that they would

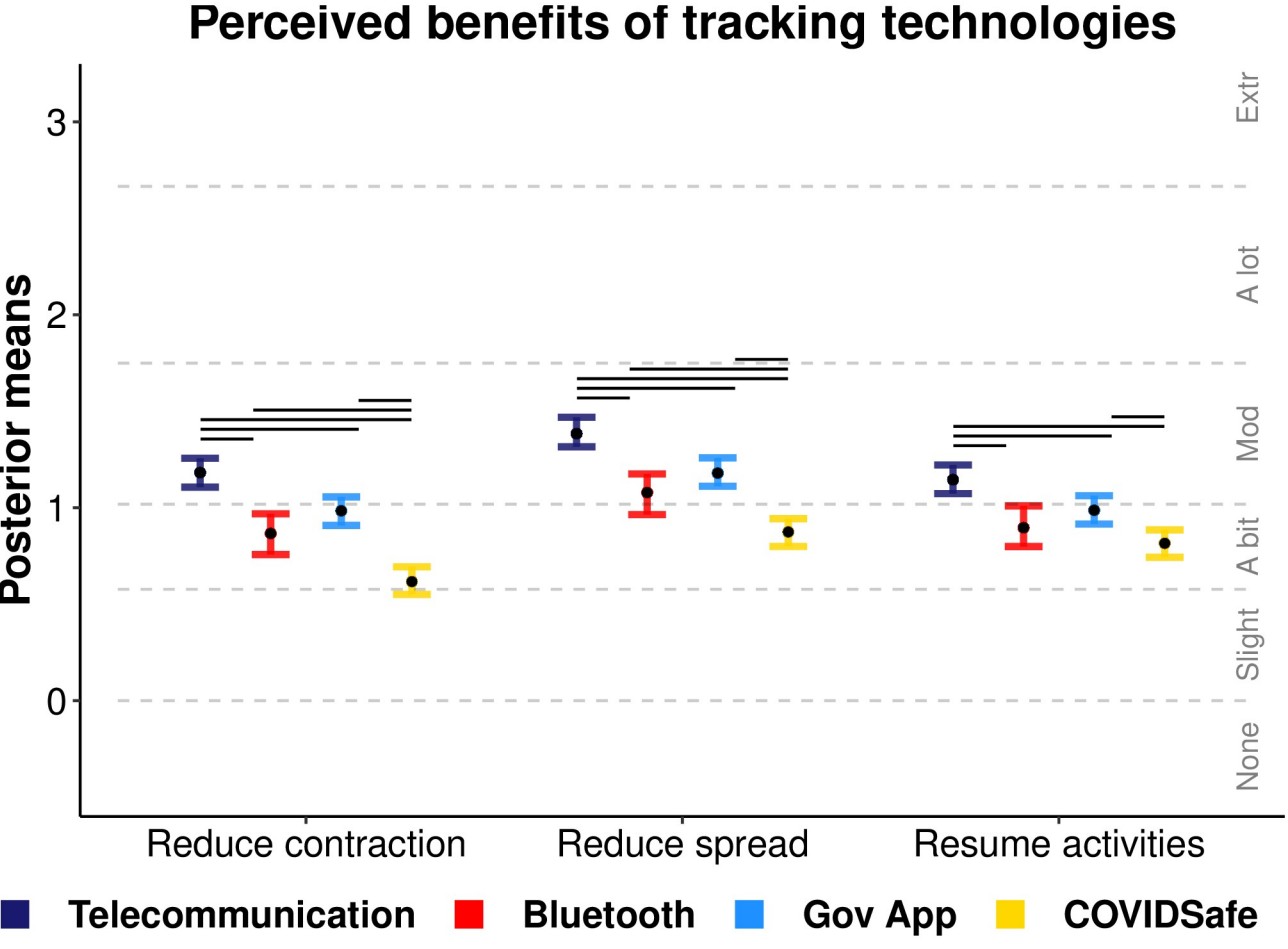

**Fig 5. Perceived benefits regression.** Ordinal regression mean posterior distributions for items assessing the perceived benefits from tracking for each scenario. Black points display mean point estimates and coloured error bars display the 95% highest posterior density interval thereof. Dotted lines indicate the ordinal regression threshold parameters which separate the continuous latent variables into the ordinal response categories made by participants ('None' to 'Extremely'). Non-overlapping intervals (within items) are denoted by horizontal bars above each comparison.

upload their registry data if they tested positive for COVID-19, assisting manual contact tracers.

App users reported their phone's operating system as primarily Android (53%) or Apple iOS (46%). Prior to June 19th 2020, COVIDSafe had a known issue on Apple iOS devices requiring users to leave their phone screen unlocked, Bluetooth on, and COVIDSafe open on the front screen to function effectively. Forty-five percent of iOS users indicated they used COVIDSafe in this fashion. Inferring from this proportion, we estimate COVIDSafe effective usage across all smartphone operating systems prior to June 19th as approximately 61%.

**App sharing and future usage.**   Fifty-six percent of users had tried to share COVIDSafe with friends and family. The app has an inbuilt sharing feature that can invite others to download the app. By Sample 4, 32% of app users knew of this functionality, 19% had used this feature, and 61% intended to do so in the future. Across both app users and non-app users, participants estimated that 33% (SD = 27%) of the people they know have downloaded the app. This proportion increased to 45% (SD = 28%) among app users and decreased to 23%

## Perceived risks of tracking technologies

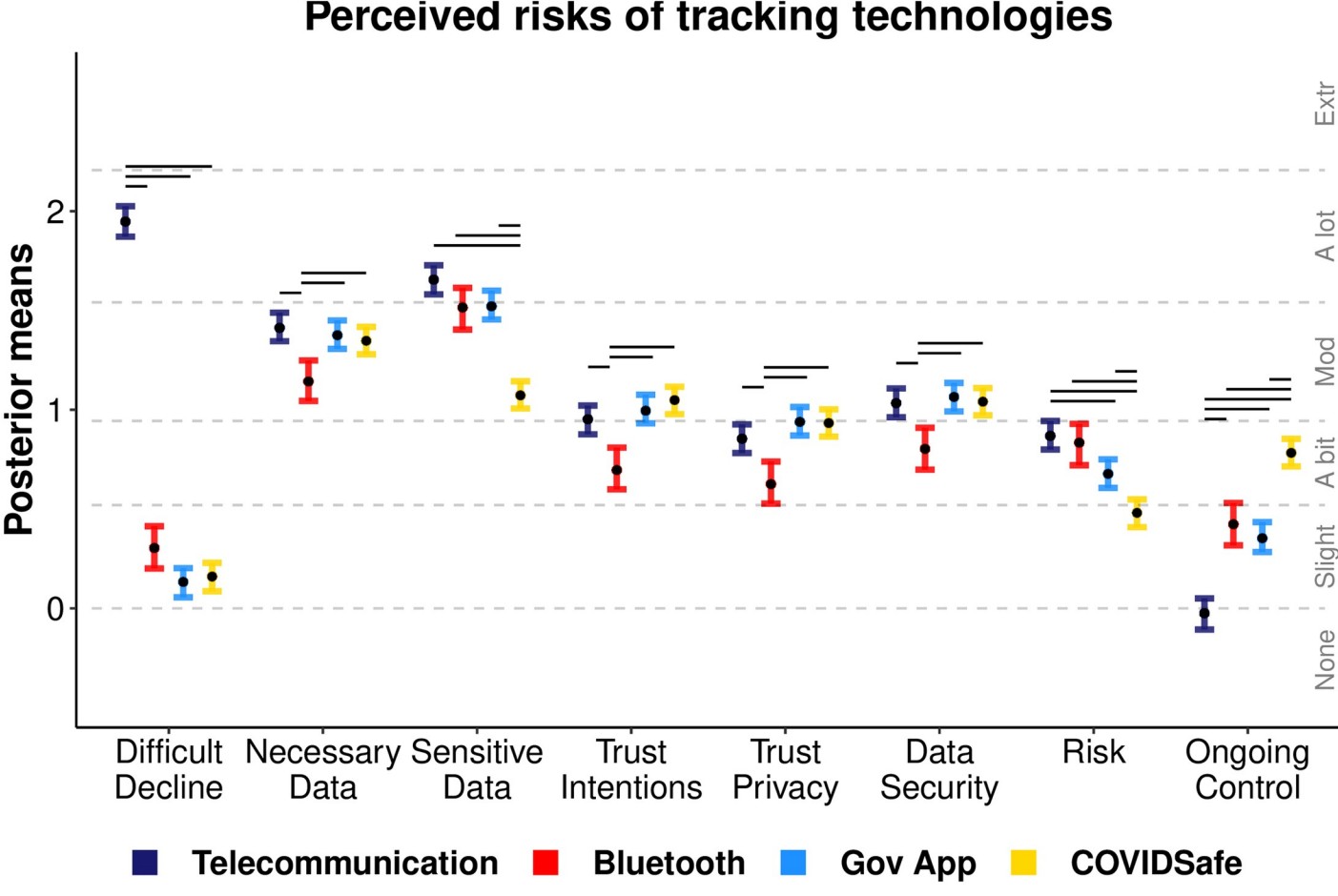

**Fig 6. Privacy perceptions regression.** Ordinal regression mean posterior distributions for items assessing the privacy perceptions of tracking for each scenario. Black points display mean point estimates and coloured error bars display the 95% highest posterior density interval thereof. Dotted lines indicate the ordinal regression threshold parameters which separate the continuous latent variables into the ordinal response categories made by participants ('None' to 'Extremely'). Non-overlapping intervals (within items) are denoted by horizontal bars above each comparison.

(SD = 21%) among non-app users. Across all participants, it was estimated that 47% (SD = 22%) of the Australian population will download and use the COVIDSafe app.

**App requirements for travel.** COVID-19 tracking apps, such as COVIDSafe, may become mandatory when entering a country. In Sample 4 of data collection, we asked participants about their attitudes towards downloading another country's app, and to requiring travellers to download COVIDSafe. When probed, 54% of participants somewhat-to-strongly agreed with downloading another country's COVID-19 tracking app if it were a condition of entry to that country; 29% somewhat-to-strongly disagreed, and 15% neither agreed or disagreed. Sixty-seven percent of participants somewhat-to-strongly agreed that all arrivals to Australia should be required to download and use COVIDSafe as a condition of entry; 14% somewhat-to-strongly disagreed, and 14% neither agreed or disagreed. This shows an imbalance in which 13% more participants would agree to impose COVIDSafe on people arriving in Australia, but would not be willing to use a similar tracking app if they were to travel to another country.

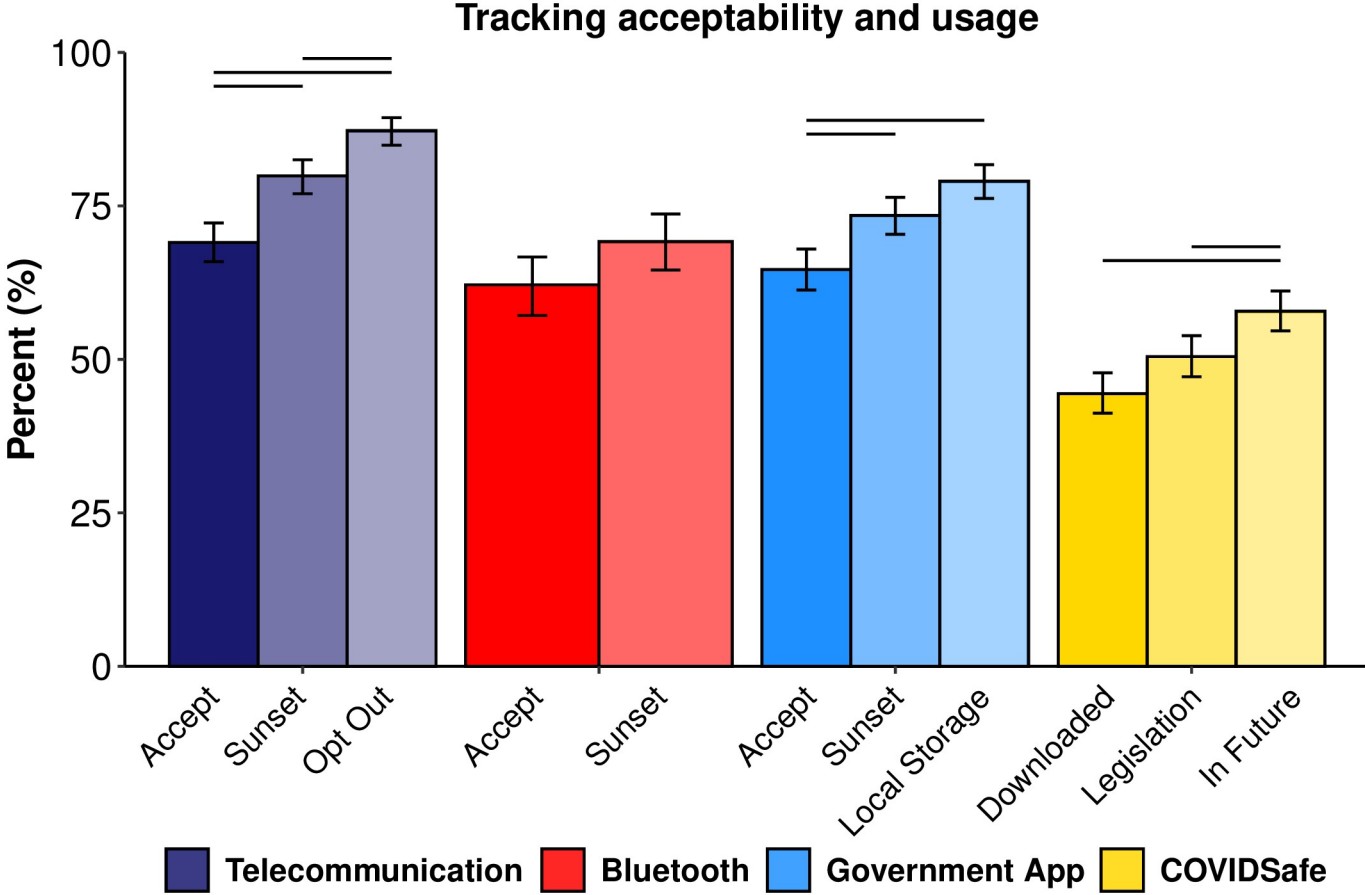

**Fig 7. Perceived acceptability and uptake.** Acceptability of each tracking scenario collapsed across samples under various conditions. Error bars are 95% Bayesian credible intervals and non-overlapping intervals (within tracking scenario) are denoted by horizontal bars above each comparison.

**Perceptions of the COVIDSafe technology.** Fig 9 displays public perceptions of what technology COVIDSafe uses, among those who have and have not downloaded the app. Error bars display 95% HDIs. A larger proportion of app users identified COVIDSafe as using Bluetooth technology (70%) when compared to non-app users (39%). A small proportion of app users thought COVIDSafe used location, telecommunication or an unknown technology. An equivalent percentage of non-app users perceived COVIDSafe to use location or an unknown tracking technology (28–29%).

**Reasons for (not) downloading COVIDSafe.** COVIDSafe users were asked to arrange a series of options from most- to least-important describing i) why they downloaded the COVIDSafe app, and ii) who convinced them to download the app. A breakdown of these results are provided in the supporting information (see S1 and S2a Figs). COVIDSafe was primarily downloaded for reasons of government policy (29%) and concern over other's health (25%). Overall, self-health and a return to normal activities were responded to most frequently within the first three rank-positions. The economy was the least-important reason for downloading the app (ranked last by 46% of users). Politicians, government advertising and the Prime Minister primarily convinced users to download the app, however, other public figures, scientists and 'I did not need convincing' were most frequently ranked within the top-three options.

## Tracking acceptability and future intentions

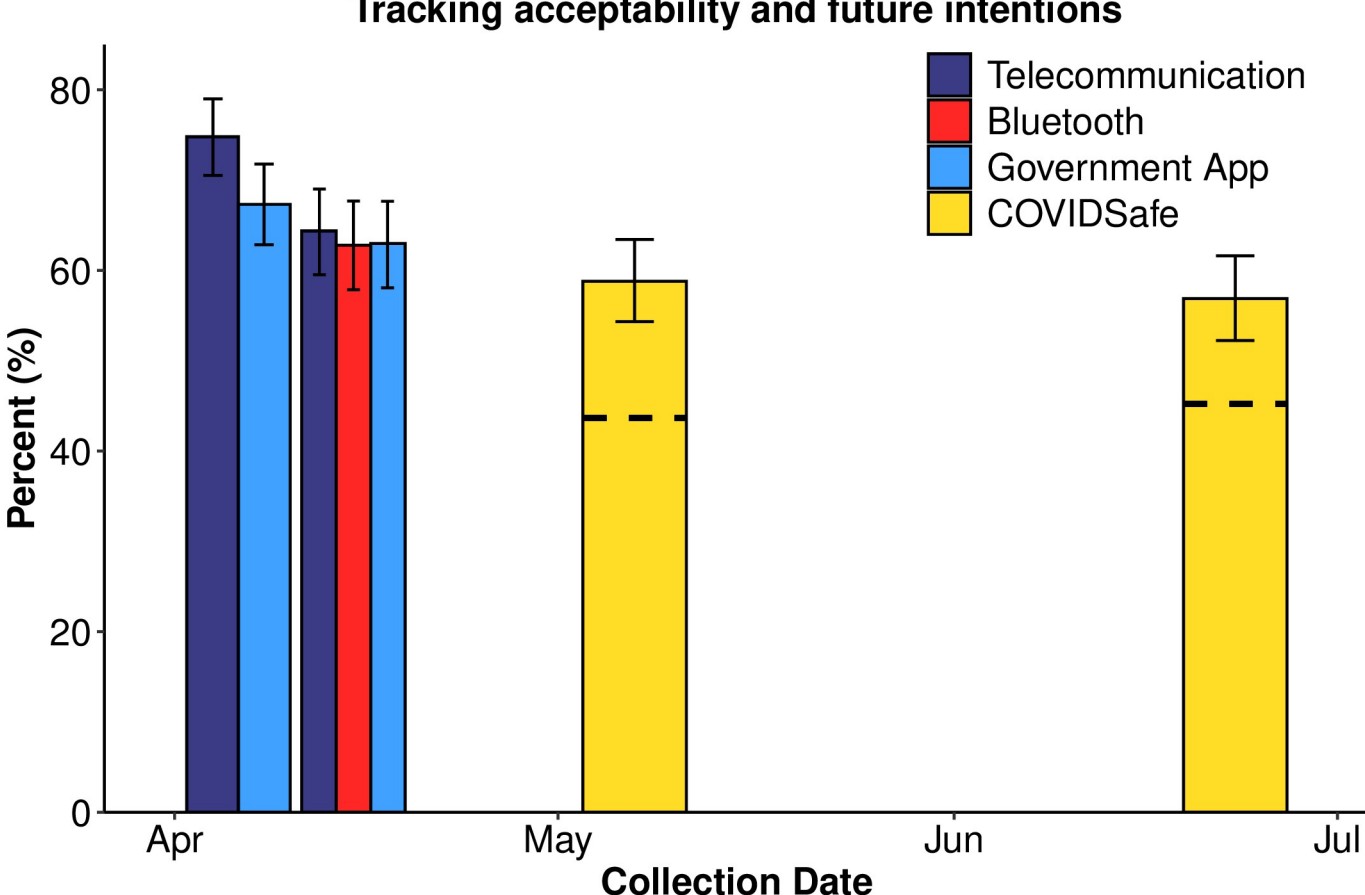

**Fig 8. Perceived acceptability and uptake by date.** Tracking acceptability and COVIDSafe downloads plotted by the date of data collection. COVIDSafe results are displayed for current app downloads (dashed line) and future app downloads (whole-bar). Error bars are 95% Bayesian credible intervals.

We asked non app users to similarly arrange options for who convinced them not to download COVIDSafe. Non-users primarily reported that they 'did not need convincing' (25%) or were convinced by other public figures (17%) and sciences (16%; see supporting information S2b Fig). The Prime Minister was ranked as the last option by 82% of respondents.

Participants who intend to download COVIDSafe in the future were asked to arrange reasons for why they are yet to download the app (supporting information, S3a and S3b Fig). Participants were primarily weighing the pros and cons, 22% and 28% in Sample 3 and 4 respectively. When the app was newly released (Sample 3), time, technological issues, and not leaving the house were among the top three responses; however, these were surpassed by 'waiting on others' and 'not considered it yet' by Sample 4. Participants who never intend to download COVIDSafe arranged options to describe their reasoning (supporting information, S3c and S3d Fig). In Samples 3 and 4 respectively, the primary reasons were privacy (20–30%), battery usage (16–19%), Government trust (14–18%), and a belief that the app will not be effective (14–19%). In Sample 4 only, participants also reported concerns about normalizing Government tracking (16%).

# Perceived COVIDSafe tracking technology

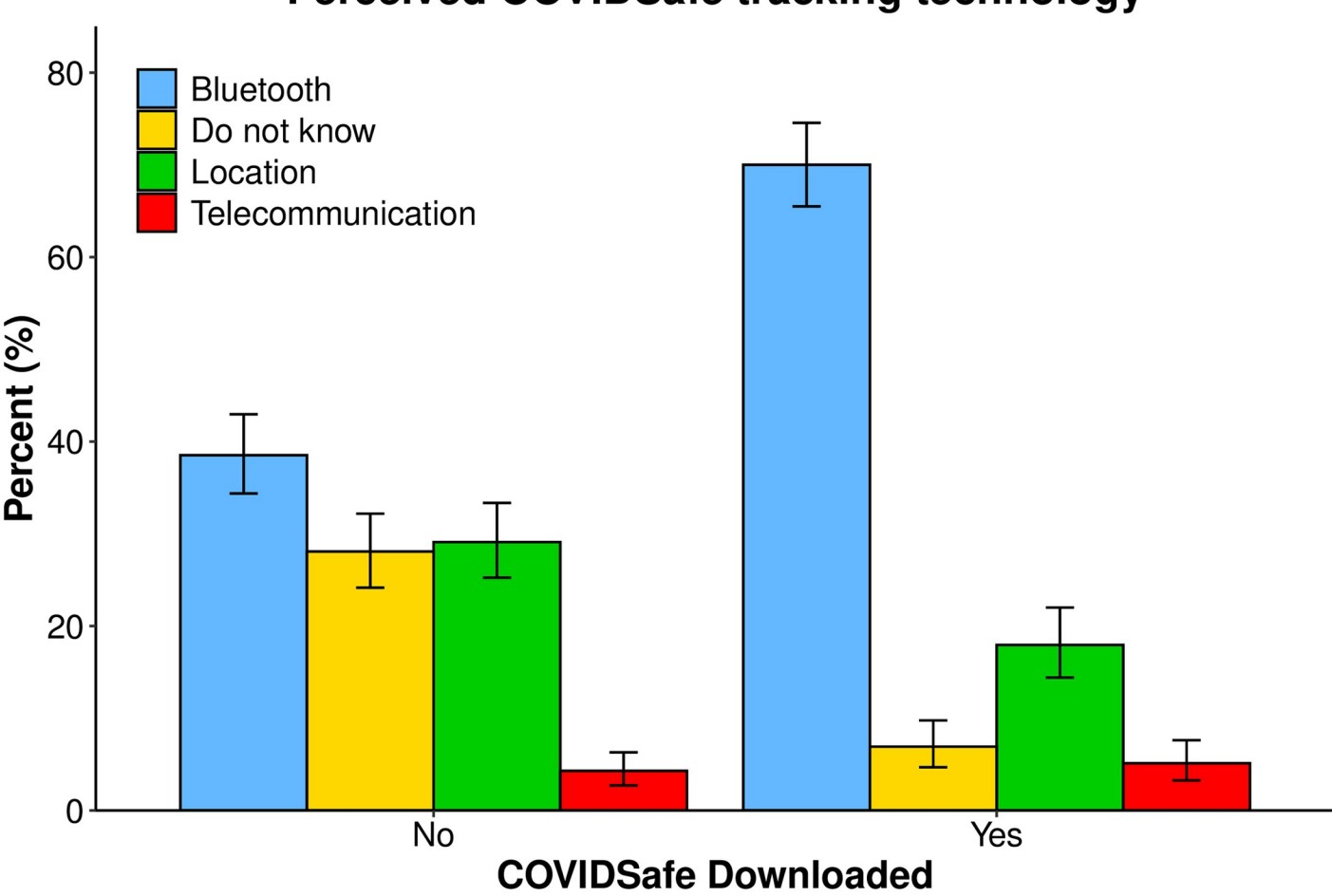

**Fig 9. Perceived COVIDSafe technology.** Public perceptions of the tracking technology used by the COVIDSafe app, grouped by whether participants reported having downloaded it. Error bars are 95% Bayesian credible intervals. Participants who had downloaded the app were much more likely to report the correct technology used by the app: Bluetooth tracking.

## Discussion

In April 2020, we asked two representative samples of the Australian public to rate the perceived risks posed by COVID-19, before rating the acceptability, effectiveness, benefits and risks posed by three hypothetical COVID-19 tracking scenarios: telecommunication network tracking, a decentralized Bluetooth Exposure Notification (EN) system backed by Apple and Google, and a partially centralized government app. Results were compared to two Australian representative samples collected in May and June on their usage and attitudes towards a real-world centralized government app, COVIDSafe.

Perceived risks from COVID-19 decreased over the collection dates with the decline of newly reported cases. Acceptability for the hypothetical tracking scenarios was moderate-to-high (62–70%) and did not vary across the scenarios; however, these attitudes did not translate to behaviour. Downloads and intentions to use COVIDSafe were lower than acceptance for the (very similar) hypothetical government app, highlighting an intention-behavior gap. Acceptability of the hypothetical tracking scenarios and COVIDSafe uptake did not vary across

the collection dates, except for telecommunication tracking, in which acceptability decreased between the first and second samples. Overall, COVIDSafe usage was motivated by government policy and a desire to safeguard public and personal health, and to return to normal activities sooner. Those who did not download the app were motivated by concerns over privacy, a lack of government trust, battery usage, and a fear of normalizing government tracking. The following discussion highlights our key findings and aims to inform policy makers.

### Hypothetical tracking scenarios

No one scenario was perceived to be more risky than another and mandatory telecommunication tracking was generally perceived to be the most beneficial technology. Data privacy, security, and trustworthiness were perceived as lower for Apple and Google's EN system than for the comparable government app, and telecommunication tracking was perceived as more difficult to decline and as having less ongoing control over the data than the other hypothetical scenarios. These differences reflect the different dimensions on which each scenario varied (e.g., voluntary vs mandatory uptake, centralized vs decentralized storage). However, regardless of how the hypothetical scenarios varied in their risk-benefit profiles, baseline acceptance remained consistently high. This indicates that these factors trade-off as a consequence of each individual's internal privacy calculus. Identifying the degree to which each of these factors contribute to this privacy calculus (à la [38]) is beyond the scope of the current study, however, represents an important line of future research.

The results of the current study have clear implications for potential corporate and government policy decisions. Historically, Australians have disproportionately trusted government agencies (84%) over private companies like Google and Apple (42%) [39], and only accepted surveillance if it was deemed necessary [15]. Contrasting these established norms, our results imply that Australians are willing to sacrifice privacy to governments and corporations in order to combat the spread of COVID-19. This being said, acceptance increased in all hypothetical scenarios with the introduction of additional privacy preserving measures (e.g., a Sunset clause) implying that privacy remains a key issue among the Australian public.

### The intention-behavior gap in government apps

Acceptability for the hypothetical centralized government app did not reflect real-world usage of the COVIDSafe app. In our samples, government trust was consistently "moderate" and the perceived risks from tracking were equivalent (data privacy and security) or lower (data risk and sensitivity) for COVIDSafe than for the hypothetical government app. COVIDSafe was also perceived as having the most ongoing control with regards to participant's data. These benefits would suggest that downloads should be greater for COVIDSafe than indicated by acceptance for the hypothetical government app.

However, the perceived harm and personal risk posed by COVID-19 and the perceived benefits from COVID-19 tracking decreased by the time of the COVIDSafe Samples. This plays an important role in the 'privacy calculus' performed by the Australian public. Most app downloads occurred within the first week of the app's launch when cases were decreasing (Fig 1), and remained stable thereafter. It appears that while cases are low, otherwise accepting members of the Australian public may perceive COVID-19 tracking to be unnecessary, explaining the observed intention-behaviour gap.

### Reasons for [not] downloading COVIDSafe

Nearly one-third of participants reported government policy as the primary reason for downloading COVIDSafe and 56% reported government officials and advertising as providing the

most convincing arguments to download the app. By contrast, one-quarter of those who did not (and will not) download the app did not need convincing to reach this decision. These individuals were primarily concerned by issues of data security (e.g., preserving privacy, normalizing government tracking, and a belief that the government was not trustworthy) and functionality (e.g., battery drain and a belief the app would not be effective), and were disproportionately misinformed about the COVIDSafe tracking technology.

Individuals who intend to download the app in the future were still weighing the pros and cons of doing so, or waiting on others to download the app for it to be effective. With only 44% of our sample downloading COVIDSafe and an effective usage rate between 61–87% (pre vs. post the June 19th 2020 iOS update), convincing this undecided cohort is critical to improving uptake. We observed other people's health, returning to normal activities, and safeguarding one's own health were primary reasons for downloading COVIDSafe. Furthermore, concern for others' health did not diminish as COVID-19 cases declined. We surmise that these are critical factors when encouraging tracking technology uptake in undecided members of the public.

## Limitations

Our study only included three hypothetical tracking scenarios, a relatively small sample of the potential technological solutions now available [40]. Furthermore, the hypothetical scenarios presented in our surveys do not perfectly describe real-world policies or technologies that have been implemented. The hypothetical Bluetooth scenario and the government app scenario were both created before the Apple and Google EN system and COVIDSafe app were developed and are not perfect descriptions of these technologies. However, it seems unlikely that minor inconsistencies in our hypothetical scenarios and the real-world tracking technologies would drastically alter public perceptions. Our measures were also made when COVID-19 cases were declining, altering how policy makers need to interpret the results (see the following section on policy implications).

The direct comparison of technology acceptance is also limited by differences among the scenario descriptions. For example, the telecommunication tracking scenario was accompanied by mandatory fines or arrests, while the other scenarios were not. Although these differences reflect real-world policy decisions—fines may accompany mandatory tracking but not a voluntary app—readers must be aware that reported acceptance is more than simply whether the technology itself was acceptable, but instead, whether it was acceptable within a specific context. Policy makers should bear this in mind when comparing acceptability ratings between the scenarios. Finally, our samples may have an inherent bias towards technology acceptance due to our assessment of technological issues through an online survey.

## Policy implications

Public acceptance for tracking technologies when COVID-19 cases were declining (see Fig 1) did not translate to app uptake when cases were close to zero. Our results show that acceptability and app take is dependent upon i) the privacy preserving measures that accompany the technology, and ii) the perceived severity of the pandemic. Therefore, policy makers may view our results as a benchmark of Australian attitudes and behaviors when cases are low. Indeed, our results suggest that uptake of COVIDSafe and acceptance for tracking technologies will only increase if COVID-19 cases were to again increase in Australia.

Our results suggest that the Australian Government has broad support to introduce privacy encroaching tracking technologies as long as they are perceived to be effective and necessary. Our results indicate policymakers can increase acceptance and uptake of tracking technologies

in two ways. First, policymakers could leverage privacy calculus by improving the messaging and information dissemination regarding the privacy protections that accompany tracking technologies. Second, policymakers could focus on factors unrelated to privacy calculus, for example, 'nudge' factors [41], that aim to make the technology more usable, and easier to communicate and share with others. We surmise that targeting these factors may be more beneficial to acceptance and uptake than modifying the technical details of the technology.

Finally, our analysis of rank-ordered reasons for (not) downloading COVIDSafe provides a clear framework to deliver policy decisions. Arguments for tracking technologies appear most convincing when presented by trusted government individuals and advertisers, and most effective when the messaging focuses on i) other people's health, ii) returning life to normal activities, and iii) safe guarding one's own health.

## Conclusion

The COVID-19 pandemic is one of the largest health crises of the past century, and in lieu of a vaccine or treatment, requires new technological solutions to stop its spread and return societies to normal life. We found acceptability was high among the Australian public for three hypothetical privacy-encroaching COVID-19 tracking technologies—telecommunication network tracking, the Apple/Google exposure notification system, and a centralized government app. We also observed that acceptability may not translate to public uptake. COVIDSafe downloads were far fewer than predicted by the rate at which participants endorsed the hypothetical government app, and appeared to be impacted by privacy concerns and the risk of infection. Inferring from our results, we provide clear policy implications to assist decision makers in introducing effective methods of COVID-19 tracking. This study, conducted on a representative Australian sample, can inform decision makers the world over in their COVID-19 pandemic responses, particularly in countries with similar cultures and attitudes towards privacy and public health.

## Supporting information

**S1 Fig. Rank ordered responses for why people downloaded the COVIDSafe app.** Participants were asked to rank order from most-important to least-important the five reasons presented on the y-axis of S1 Fig. The cumulative responses probabilities for each item are displayed in blue and the percentage of responses at each ranked-position are displayed by text in each cell, for example, in the first ranked-position Government policy was viewed as most important (29%) followed by others health (25%). Returning to normal activities was most frequently ranked in the second position (38%) followed by self health (28%). To show which items were viewed as most important overall, the cumulative percentage of responses to each item are shaded in blue, with cumulative percentages increasing from left-to-right (lighter to darker). For example, return to normal activities, self health, and others health were perceived as the important reasons overall, accruing the most responses in the first three rank-positions. Overall, the economy was ranked the least-important reason for down-loading COVIDSafe. (TIF)

**S2 Fig. Rank ordered responses for who convinced participants to download the COVIDSafe app (a) or who convinced participants to not download the app (b).** Each plot displays data collapsed across Samples 3 and 4; response percentages are displayed in text and the cumulative responses probabilities for each item are displayed in blue. When asked, most participants responded that other politicians, government advertising and the Prime Minister were the primary individuals who convinced them to download COVID Safe (S2a Fig). Public

figures, scientists and did not need convincing were frequently ranked in the first three-positions. The Prime Minister was ranked as least important by 50% of respondents. Most participants who did not download COVIDSafe indicated they 'did not need convincing' or were convinced by other public figures and scientists (S2b Fig). Friends and family were frequently ranked in the top-three positions. The Prime Minister was ranked least important by 82% of respondents.
(TIF)

**S3 Fig.** Rank ordered responses asking individuals why they are yet to download the COVID-Safe app in Samples 3 (a) and 4 (b), or why they will never download the app in Samples 3 (c) and 4 (d). Response percentages are displayed in text and cumulative responses probabilities for each item are displayed in blue. 'Weighing the pros and cons' was the primary reason for not yet downloading the app. In Sample 3 when the app was newly released, time, technical issues and not leaving the house were among the top reasons, however, were replaced in Sample 4 with less time-sensitive issues: 'waiting on others' and 'not considered it'. In both samples, concerns regarding privacy, battery usage, Government trust, and a belief that the app will be non effective were primary reasons for never downloading COVIDSafe. In Sample 4, concern regarding the normalizing of Government tracking became a primary issue.
(TIF)

## Author Contributions

**Conceptualization:** Paul M. Garrett, Joshua P. White, Stephan Lewandowsky, Yoshihisa Kashima, Andrew Perfors, Daniel R. Little, Nic Geard, Lewis Mitchell, Martin Tomko, Simon Dennis.

**Data curation:** Paul M. Garrett.

**Formal analysis:** Paul M. Garrett, Joshua P. White, Simon Dennis.

**Funding acquisition:** Joshua P. White, Simon Dennis.

**Investigation:** Paul M. Garrett, Joshua P. White, Stephan Lewandowsky, Yoshihisa Kashima, Andrew Perfors, Daniel R. Little, Nic Geard, Lewis Mitchell, Martin Tomko, Simon Dennis.

**Methodology:** Paul M. Garrett, Joshua P. White, Stephan Lewandowsky, Yoshihisa Kashima, Andrew Perfors, Daniel R. Little, Nic Geard, Lewis Mitchell, Martin Tomko, Simon Dennis.

**Project administration:** Paul M. Garrett, Joshua P. White, Simon Dennis.

**Resources:** Simon Dennis.

**Software:** Paul M. Garrett, Joshua P. White.

**Supervision:** Yoshihisa Kashima, Andrew Perfors, Simon Dennis.

**Validation:** Paul M. Garrett, Joshua P. White.

**Visualization:** Paul M. Garrett, Joshua P. White, Simon Dennis.

**Writing – original draft:** Paul M. Garrett, Joshua P. White, Simon Dennis.

**Writing – review & editing:** Paul M. Garrett, Joshua P. White, Stephan Lewandowsky, Yoshihisa Kashima, Andrew Perfors, Daniel R. Little, Nic Geard, Lewis Mitchell, Martin Tomko, Simon Dennis.

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
