## [Decision Letter · Decision Letter 0]

4 Nov 2020

PONE-D-20-30595

The acceptability and uptake of smartphone tracking for COVID-19 in Australia

PLOS ONE

Dear Dr. Garrett,

Thank you for submitting your manuscript to PLOS ONE. After careful consideration, we feel that it has merit but does not fully meet PLOS ONE’s publication criteria as it currently stands. Therefore, we invite you to submit a revised version of the manuscript that addresses the points raised during the review process.

Please find below the reviewers' comments, as well as those of mine.

We look forward to receiving your revised manuscript.

Kind regards,

Valerio Capraro

Academic Editor

PLOS ONE

Journal Requirements:

Additional Editor Comments (if provided):

I have now collected two reviews from two experts in the field. The reviewers like the paper and think that it can be published after a suitable revision. Therefore, I would like to invite you to revise your paper following the reviewers' comments. Moreover, I would like to add one minor comment. I have noticed that the literature review contains several unpublished papers. I think that references to unpublished works should be, if possible, minimised. So, my suggestion is to replace, when possible, unpublished works with published works. A useful starting point could be the "perspective article" on what social and behavioral science can do to support pandemic response (Van Bavel et al. 2020).

Looking forward for the revision.

References

Van Bavel, J. J., et al. (2020). Using social and behavioural science to support COVID-19 pandemic response. Nature Human Behaviour, 1-12.

Reviewers' comments:

Reviewer's Responses to Questions

**Comments to the Author**

1. Is the manuscript technically sound, and do the data support the conclusions?

Reviewer #1: Yes

Reviewer #2: Yes

2. Has the statistical analysis been performed appropriately and rigorously? 

Reviewer #1: Yes

Reviewer #2: Yes

3. Have the authors made all data underlying the findings in their manuscript fully available?

Reviewer #1: Yes

Reviewer #2: Yes

4. Is the manuscript presented in an intelligible fashion and written in standard English?

Reviewer #1: Yes

Reviewer #2: Yes

5. Review Comments to the Author

Reviewer #1: The authors report the results of several surveys that were carried out to gauge the attitude of the Australian population towards tracking technologies. The paper is scientifically sound, well written, easy to follow and the results are quite interesting. I believe the results have important policy implications, especially the ones regarding the intention-behavior gap.

By the way, I found a small spelling mistake:

line 491: upp -> app

Reviewer #2: The article is timely and deals with a very important issue. However, there are several weak points that I listed below:

1. Studies presented in this paper seem to be descriptive to a large extent. The Introduction is lacking with any hypotheses and the theoretical justification for the usage of three types of scenarios appears insufficient. I suggest authors elaborate more on the mechanism that may stand behind the attitudes towards different tracking technologies, even if they do not have firm hypotheses. There is a recent paper by Wnuk, A., Oleksy, T., & Maison, D. (2020). The acceptance of Covid-19 tracking technologies: The role of perceived threat, lack of control, and ideological beliefs. PloS one, 15(9) that shows that acceptance of tracking technologies is predicted by perceived risk of Covid-19 infection, endorsement of liberty and RWA. It seems to be a relevant background for this paper.

2. The three scenarios that were used differ in many points, not only in the source of tracking apps (government vs telecommunication vs Bluetooth). For example, in the second scenario there is information that “(…)Australian Government who may use the data to locate people who were violating lockdown orders and enforce them with fines and arrests where necessary.” This makes it difficult to say what was the actual reason people thought it would be beneficial, i.e. whether because the app was mandatory or because the government would impose additional restrictions on those who did not follow the rules. This is the main limitation of this study.

3. In the Discussion I would recommend discussing the possible reasons that may cause the results, e.g. why there were no differences between scenarios in terms of acceptance of technologies.

6. PLOS authors have the option to publish the peer review history of their article (what does this mean?). If published, this will include your full peer review and any attached files.

Reviewer #1: No

Reviewer #2: No

---

## [Author Response · Author response to Decision Letter 0]

6 Dec 2020

The following is a copy of the attached responses letter.

Response to reviewers

I begin by thanking the Editor and both Reviewers for their excellent feedback. The comments provided were very constructive and the manuscript was returned in a rapid timeframe. This is a remarkable effort on the part of all parties under normal circumstances, let alone during the current pandemic. Thank you.

Below I detail my replies to each issue raised. For clarity of exposition, reviewer or editor comments are displayed in black, replies are in blue, and modified text in the pdf is displayed in red. Manuscript line numbers of each modification are also reported in this reply letter.

Again, thank you for your time and feedback on this manuscript. We hope these changes are acceptable to the editor and reviewers, and we look forward to seeing our work published in PloS One.

Sincerely,

Dr Paul M. Garrett

Editor

I have now collected two reviews from two experts in the field. The reviewers like the paper and think that it can be published after a suitable revision. Therefore, I would like to invite you to revise your paper following the reviewers' comments. Moreover, I would like to add one minor comment. I have noticed that the literature review contains several unpublished papers. I think that references to unpublished works should be, if possible, minimised. So, my suggestion is to replace, when possible, unpublished works with published works. A useful starting point could be the "perspective article" on what social and behavioral science can do to support pandemic response (Van Bavel et al. 2020).

 - Thank you for this feedback and the suggested reference; this has been added to the manuscript. 

 - Upon resubmission, all previously cited journal articles have been accepted for publication and our references have been updated accordingly. We have included two additional preprints by co-authors of this paper that are relevant to the current study, Garrett et al., (2020) and Lewandowsky et al., (2020), which use similar methods to determine attitudes to tracking technologies in Taiwan and the United Kingdom, respectively. We denote these as ‘Preprints’ in our reference list.

- The authors believe that the necessary changes have been made to align this manuscript with Plos One’s formatting guides. We will of course change anything else upon request. Thank you.

Reviewer 1

The authors report the results of several surveys that were carried out to gauge the attitude of the Australian population towards tracking technologies. The paper is scientifically sound, well written, easy to follow and the results are quite interesting. I believe the results have important policy implications, especially the ones regarding the intention-behavior gap.

line 491: upp -> app 

 - Done, thank you. 

Reviewer 2

The article is timely and deals with a very important issue. However, there are several weak points that I listed below:

1. Studies presented in this paper seem to be descriptive to a large extent. The Introduction is lacking with any hypotheses and the theoretical justification for the usage of three types of scenarios appears insufficient. I suggest authors elaborate more on the mechanism that may stand behind the attitudes towards different tracking technologies, even if they do not have firm hypotheses. There is a recent paper by Wnuk, A., Oleksy, T., & Maison, D. (2020). The acceptance of Covid-19 tracking technologies: The role of perceived threat, lack of control, and ideological beliefs. PloS one, 15(9) that shows that acceptance of tracking technologies is predicted by perceived risk of Covid-19 infection, endorsement of liberty and RWA. It seems to be a relevant background for this paper.

“I suggest authors elaborate more on the mechanism that may stand behind the attitudes towards different tracking technologies…”

 - Done, thank you. Following our introduction of the privacy calculus mechanism, we have added further content on the elements that may inform tracking acceptance (lines 28 – 38).

“Theoretical justification for the usage of three types of scenarios appears insufficient…”

 - Done, thank you. Our investigation was informed by three real-world policy options that have been introduced around the world. We now make this real-world motivation clear in the introduction (lines 40 – 47).

“The Introduction is lacking with any hypotheses…”

 - Done, thank you. We chose to not hypothesise about public acceptance, primarily because each scenario differed on multiple dimensions (e.g., voluntary vs mandatory tracking, centralized vs decentralized storage). We still believe it more appropriate to simply report the findings as a descriptive account of public attitudes, and therefore have chosen not to add hypothesises after the results are known (HARKing; see Kerr, 1998; Rubin, 2017). We now make this clear to the reader (lines 97 – 105). 

2. The three scenarios that were used differ in many points, not only in the source of tracking apps (government vs telecommunication vs Bluetooth). For example, in the second scenario there is information that “(…)Australian Government who may use the data to locate people who were violating lockdown orders and enforce them with fines and arrests where necessary.” This makes it difficult to say what was the actual reason people thought it would be beneficial, i.e. whether because the app was mandatory or because the government would impose additional restrictions on those who did not follow the rules. This is the main limitation of this study.

 - Done, thank you. We have included this limitation in our discussion (lines 516 – 523).

3. In the Discussion I would recommend discussing the possible reasons that may cause the results, e.g. why there were no differences between scenarios in terms of acceptance of technologies.

 - Done, thank you. We have expanded our discussion under the subsection “Hypothetical tracking scenarios” to this effect (lines 452 – 459).

References

Garrett, P., Wang, Y. W., White, J. P., Hsieh, S., Strong, C., Lee, Y. C., ... & Dennis, S. (2020). Young adults view smartphone tracking technologies for COVID-19 as acceptable: the case of Taiwan. (https://psyarxiv.com/mznrb) [Preprint; under review at Psychology, Public Policy and Law]

Kerr, N. L. (1998). HARKing: Hypothesizing after the results are known. Personality and social psychology review, 2(3), 196-217.

Lewandowsky, S., Dennis, S., Perfors, A., Kashima, Y., White, J. P., Garrett, P., ... & Yesilada, M. (2020). Public acceptance of privacy-encroaching policies to address the covid-19 pandemic in the united kingdom. (https://psyarxiv.com/njwmp) [Preprint; revised & resubmitted to Plos One]

Rubin, M. (2017). When does HARKing hurt? Identifying when different types of undisclosed post hoc hypothesizing harm scientific progress. Review of General Psychology, 21(4), 308-320.

---

## [Decision Letter · Decision Letter 1]

17 Dec 2020

The acceptability and uptake of smartphone tracking for COVID-19 in Australia

PONE-D-20-30595R1

Dear Dr. Garrett,

We’re pleased to inform you that your manuscript has been judged scientifically suitable for publication and will be formally accepted for publication once it meets all outstanding technical requirements.

Kind regards,

Valerio Capraro

Academic Editor

PLOS ONE

Additional Editor Comments (optional):

Reviewers' comments:

Reviewer's Responses to Questions

**Comments to the Author**

1. If the authors have adequately addressed your comments raised in a previous round of review and you feel that this manuscript is now acceptable for publication, you may indicate that here to bypass the “Comments to the Author” section, enter your conflict of interest statement in the “Confidential to Editor” section, and submit your "Accept" recommendation.

Reviewer #2: All comments have been addressed

2. Is the manuscript technically sound, and do the data support the conclusions?

Reviewer #2: Yes

3. Has the statistical analysis been performed appropriately and rigorously? 

Reviewer #2: Yes

4. Have the authors made all data underlying the findings in their manuscript fully available?

Reviewer #2: Yes

5. Is the manuscript presented in an intelligible fashion and written in standard English?

Reviewer #2: Yes

6. Review Comments to the Author

Reviewer #2: Thank you for your edits and clarifications. I find the manuscript improved. Good luck with the publication!

7. PLOS authors have the option to publish the peer review history of their article (what does this mean?). If published, this will include your full peer review and any attached files.

Reviewer #2: No

---

## [Editor Report · Acceptance letter]

21 Dec 2020

PONE-D-20-30595R1 

The acceptability and uptake of smartphone tracking for COVID-19 in Australia 

Dear Dr. Garrett:

I'm pleased to inform you that your manuscript has been deemed suitable for publication in PLOS ONE. Congratulations! Your manuscript is now with our production department. 

Kind regards, 

on behalf of

Dr. Valerio Capraro 

Academic Editor

PLOS ONE